# Identification and safety assessment of *Enterococcus thailandicus* TC1 isolated from healthy pigs

Xiaoying Wu[1☯], Bei Wu[2☯], Yue Li[2], Xiue Jin[3], Xiliang Wang[2]*

**1** Department of Biology, Taiyuan Normal College, Taiyuan, PR China, **2** State Key Laboratory of Agricultural Microbiology, College of Veterinary Medicine, Huazhong Agricultural University, Wuhan, P.R. China, **3** Hubei Provincial Institute of Veterinary Drug Control, Wuhan, P.R. China

☯ These authors contributed equally to this work.
* wxl070@mail.hzau.edu.cn

**Data Availability Statement:** All relevant data are within the manuscript and its Supporting Information files.

**Funding:** This work was supported by the National Key Research and Development Program of China

## Abstract

Enterococci have the dual characteristics of being opportunistic pathogens and promising probiotics. The isolation from patients of CDC PNS-E2, a newly described *Enterococcus* species *Enterococcus sanguinicola*, may pose potential hazards. *Enterococcus thailandicus* from fermented sausage is a senior subjective synonym of *E. sanguinicola*. In this study, *Enterococcus thailandicus* TC1 was first isolated in healthy pigs in Tongcheng, China and identified by phenotypic analysis and 16S rRNA-based techniques. To evaluate the strain safety, an approach including virulence factors, antibiotic resistance, and animal experiments was adopted. The results show that *cylA*, *gelE*, *esp*, *agg*, *ace*, *efaAfm*, *efaAfs*, *ptsD* genes were undetected, and that the strain was sensitive or poorly resistant to some clinically relevant antibiotics. However, the isolated strain demonstrated β-hemolytic activity in rabbit blood agar plates. Analysis of animal experiments revealed that the isolated strain had no adverse effect on translocation and the internal organ indices, though significant differences in histology (villi height, crypts height) of ileum were observed. The data acquired suggest that *E. thailandicus* TC1 may be associated with a potential health risk.

## Introduction

The enterococci are gram-positive catalase-negative cocci that have gone through sizable changes in systematics in the past few years. This genus has grown from the 19 species described in 2002 [1] to the 51 species described currently (http://www.bacterio.cict.fr/e/enterococcus.html). Since the *Enterococcus* was identified as separate genus [2], several new species have been isolated from clinical samples. Those isolated new species have helped to improve the identification process and have created a greater number of probiotics or opportunistic pathogens [3, 4].

Enterococci that are normal inhabitants of the gastrointestinal tracts of both humans and animals often are added to fermented foods [5, 6] or used as probiotics, which can relax or prevent several disorders such as acute gastroenteritis, antibiotic-associated diarrhea, lactose intolerance, and inflammatory bowel syndrome [7]. However, in recent years, enterococci

(2017YFD0501000) and the Fundamental Research Funds for Central Universities (2662019PY061).

**Competing interests:** The authors have declared that no competing interests exist.

infections in animals have been increasingly reported as causes of animal diarrhea, septicemia, and endocarditis [8–10], which seriously influence the development of aquaculture. The infection by *Enterococcus* can not only cause swine disease and death, but may also pose a threat to human health and, inevitably, can lead to the decrease of meat quality and increase the incidence of resistant enterococci. Therefore, safety evaluation of new enterococcal strains in probiotic preparations and food products is vital [4].

*Enterococcus* strain (CDC PNS-E1, CDC PNS-E2, and CDC PNS-E3) isolation from human clinical specimens has been reported, which indicated the association of these strains with invasive infections in humans [2, 11]. Based on the results of a multiphase taxonomic investigation, Carvalho and colleagues concluded that the unknown cocci represent new species within the genus *Enterococcus*. New Species of *Enterococcus* sp nov CDC PNS-E1, CDC PNS-E2, and CDC PNS-E3 were named by the United States (US) Center for Disease Control and Prevention (CDC) in light of the recommendation in minute 10 of the July 2002 meeting of the International Committee on Systematics of Prokaryotes Subcommittee on the taxonomy of *staphylococci* and *streptococci*, which referred to the description of a new species upon a single isolate [12]. The *enterococcus* species CDC PNS were also designated as *Enterococcus sanguinicola* sp. nov., and *E. thailandicus* from fermented sausage is a senior subjective synonym of *E. sanguinicola* [13, 14]. Although these newly discovered species have enriched the culture collection, related safety information is limited. At present, the safety of this strain has not been reported, therefore, this study aims to investigate the safety of *E. sanguinicola* TC1 isolated from healthy pigs to provide a theoretical basis for the development and application of this strain.

## Materials and methods

### Strain and vector

*Staphylococcus aureus* subsp. *aureus* Rosenbach (ATCC® 25923™) and *Enterococcus faecium* HDRsEF1 were stored in the Veterinary Microbiology and Immunology Laboratory of Huazhong Agricultural University (Wuhan, China). DH5α™ competent cells were obtained from TransGen Biotech (Beijing, China) and Pmd18-T Vector was purchased from Takara (Dalian, China).

### Reagents

Mueller Hinton (M-H) broth and de Man, Rogosa, and Sharpe (MRS) broth were purchased from Becton Dickenson (United States). Biochemical reagents (i.e. arabinose, pyruvate, tellurite, arginine, glucose, inulin, lactose, mannitol, maltose, melibiose, raffinose, ribose, sucrose, sorbitol, sorbose, trehalose, and xylose) and antimicrobial agents (i.e. ampicillin, chloramphenicol, ciprofloxacin, clarithromycin, erythromycin, gentamicin, nitrofurantoin, norfloxacin, tetracycline and vancomycin) were obtained from Hangzhou Microbe Reagent Co., Ltd. (Hangzhou, China). Primers were synthesized by the Tsingke Biotechnology Limited Company (Wuhan, China). Taq DNA Polymerase and deoxynucleotide triphosphates (dNTPs) were purchased from Takara (China). Defibrinated rabbit blood was purchased from Zhengzhou Kowloon Biological Products Co., Ltd. (Zhengzhou, China). DNA isolation kits and gel DNA purification kits were purchased from Tiangen Biotech (Beijing, China).

### Bacterial isolation

Sterile cotton swabs were used to acquire rectal content samples from 100-day-old (N = 13) and 21-day-old healthy Tongcheng pigs (N = 6) at the Tongcheng Reservation Farm and from

40-day-old healthy large white pigs (N = 11) at the Animal Husbandry Co., Ltd. Hubei Three Lake (Hubei, China). Bacterial samples were obtained from totally 30 pigs. After collection, the samples were streaked on KF-*Streptococcus* agar obtained from Hangzhou Microbe Reagent Co., Ltd. (Hangzhou, China) and incubated at 37˚C for 24–48 h. Suspected colonies, characterized by smooth red bumps surrounding apparent smooth microcolonies were selected for culture purification. Pure colonies were screened preliminarily by morphological observation, Gram staining, and peroxidase activity. Those colonies that occurred as short chains, in pairs, or singly as catalase-negative, gram-positive cocci were selected.

## Enterococcal identification

Presumptive identification at genus level was obtained by assessment of ability to grow at 45˚C and 60˚C for 30 min in broth at pH 9.6 in the presence of 6.5% (w/v) sodium chloride (NaCl) and 40% (w/v) bile, with the reaction on bile esculin agar [15].

## Phenotypic methods

Species-level strain identification was done by physiological and biochemical tests as described previously [11]. The tests performed included the following: acid production from arabinose, pyruvate utilization, gas production in MRS broth, hydrolysis of esculin in the presence of bile, tolerance to tellurite, motility, and determination of nutrient concentrations (i.e. arginine, glucose, inulin, lactose, mannitol, maltose, melibiose, raffinose, ribose, sucrose, sorbitol, sorbose, trehalose, and xylose). Each test was performed in triplicate.

## Genotypic methods

Total genomic DNA was extracted from pure cultures by a bacterial genomic DNA extraction kit according to the manufacturer's specifications. Total DNA was used as a template for 16S rRNA amplification using synthesized forward and reverse primers (Table 1). Amplified

**Table 1. PCR primers and conditions used in amplifications for the detection of virulence in *E. thailandicus* TC1.**

| Gene | Primer sequence (5'-3') | Size (bp) | Annealing temperature (˚C) |
|---|---|---|---|
| *cylA* | F: TGGATGATAGTGATAGGAAGT | 517 | 57 |
| | R: TCTACAGTAAATCTTTCGTCA | | |
| *gelE* | F: ACCCCGTATCATTGGTTTT | 419 | 52 |
| | R: ACGCATTGCTTTTCCATC | | |
| *esp* | F: TTGCTAATGCTAGTCCACGACC | 933 | 63 |
| | R: CGTCAACACTTGCATTGCCGAA | | |
| *agg* | F: AAGAAAAAGAAGTAGACCAAC | 1553 | 52 |
| | R: AAACGGCAAGACAAGTAAATA | | |
| *ace* | F: AAAGTAGAATTAGATCCACAC | 320 | 56 |
| | R: TCTATCACATTCGGTTGCG | | |
| *efaA*fs | F: GACAGACCCTCACGAATA | 705 | 52 |
| | R: AGTTCATCATGCTGTAGTA | | |
| *efaA*fm | F: AACAGATCCGCATGAATA | 735 | 52 |
| | R: CATTTCATCATCTGATAGTA | | |
| *ptsD* | F: TATCAACGCGATCAAAACGA | 241 | 52 |
| | R: CGTTCGCATACAGCTTTTCA | | |
| 16S rRNA | F:CGTGCCTAATACATGCAAGTCGAAC | 1475 | 52 |
| | R:ACGACTTCACCCCAATCATCTATCC | | |

fragments of the 16S rRNA gene were obtained, purified with a DNA clean-up kit, and sequenced by Tsingke Biotechnology Industry Co., Ltd, (Wuhan, China). Homologous sequences were queried in the Basic Local Alignment Search Tool-Nucleotide (BLASTN) data-base (National Center for Biotechnology Information, Bethesda, Maryland, USA).

### Detection of virulence genes and hemolytic phenotype assays

Enterococci virulence genes for gelatinase (*gelE*), enterococcal surface protein (*esp*), cytolysin (*cylA*), adhesion to collagen (*ace*), cell-wall adhesion (*efaA*), cell adhesion (*agg*) and *ptsD* were investigated by PCR amplification using total genomic DNA as a template. The primers used are listed in Table 1. The amplification conditions were as follows: an initial denaturation step of 94˚C for 5 min, 30 cycles of denaturation at 94˚C for 1 min, annealing for 1 min as shown in Table 1, extension at 72˚C for 1 min, and a single elongation step at 72˚C for 10 min, followed by storage at 4˚C. Hemolytic phenotype was determined by streaking enterococcal cultures on layered 5% defi-brinated rabbit blood agar plates. *Staphylococcus* ATCC 25923 was used as reference strain (posi-tive control) and *E. faecium* HDRsEF1 was used as negative control. Plates were incubated at 37˚C for 24 h [4]. When observed, the presence or absence of zones of clearing around the colonies were interpreted as β-hemolysis (positive) or γ-hemolysis (negative) activity, respectively.

### Detection of antibiotic resistance

Antimicrobial susceptibility was tested by the Kirby-Bauer disk diffusion method in M-H agar, as described by the manufacturer, and by broth microdilution according to EUCAST (www. eucast.org; version 5.0, January 2015) and results were interpreted by EUCAST or by CLSI [16]. *Staphylococcus* ATCC 25923 was used as a reference strain. The following 11 antibiotics (concentrations given in Table 3) were applied in the present study: ampicillin, chlorampheni-col, ciprofloxacin, clarithromycin, erythromycin, gentamicin, nitrofurantoin, norfloxacin, penicillin, tetracycline and vancomycin.

### Animal experiments

Twelve specific pathogen-free male Kunming mice, 6 to 8 weeks old (average weight: 29.82 ± 0.25 g; mean ± SD) were housed in a temperature-controlled environment (22 ± 2˚C, humid-ity of 56% ± 5%) with a cycle of 12 h of light and 12 h of dark throughout the experiment. All of the animals were fed ad libitum with a conventional balanced diet. After an acclimatization period of 7 days, the mice were randomly divided into control and test groups. The control group ($n$ = 6) was administered 0.2 mL 10% skim milk orally, and the test group ($n$ = 6) was administered 0.2 mL cell suspensions ($10^{10}$ cfu/mL in skim milk) of the isolated strain TC1 twice daily for 7 days. During the experiment, the animals' activity, behavior, feces, tempera-ture, and degree of hair luster were observed twice daily, and treatment-related illness or death was recorded for both the experimental group and the control group. All the animal treat-ments were carried out in accordance with the guidelines in the care and use of animals and with the approval of the Research Ethics Committee of Huazhong Agricultural University, Hubei,China.

### Bacterial translocation and internal organ indices

Following the observation period of 7 days, the animals were euthanized by cervical disloca-tion. The spleen, heart, spleen, kidney, and a sample of liver tissues were excised under strict aseptic conditions to avoid any cross-contamination. All samples were individually

homogenized with a tissue grinder. The tissue suspensions were plated separately on MRS agar plates and subsequently incubated at 37°C for 24 h under aerobic conditions [17].

Acquired internal organs including liver, heart, kidney, and spleen were weighed immediately. The organ index was expressed as the actual weight of the internal organ of each mouse divided by the last measure of live body weight [17].

### Histological assay

Small samples (0.5 cm) of the ileum (2 cm away from caecum), the caecum (middle portion), and the colon (2 cm away from caecum) were excised [18], rinsed with phosphate-buffered saline (PBS) for histological studies, and fixed in 10% neutral buffered formalin for 2 days. Tissue sections were cut at 6 μm intervals and stained with hematoxylin and eosin (H&E). The morphological parameters were measured using a micrometer under a light microscope. Each sample was measured in 10 fields for every parameter and the mean of these measurements was used for statistical analysis. Villus height was measured from the crypt-villus junction to the tip of the villus; crypt depth was measured from the base of the crypt to the crypt-villus junction [2].

### Statistical analysis

Results were expressed as means ± standard deviations (SD). Data were analyzed using the one-way analysis of variance (ANOVA) test procedure included in the SPSS version 13.0 software (SPSS Inc., Chicago, IL). Probability levels of less than .05 were considered significant.

## Results

### Isolation and identification of the strain

A total of 6 (TC1-6) strains were isolated, from red colonies situated around the medium color from purple to yellow on KF-Streptococcus agar. All of the selected colonies occurred as short chains, in pairs, or singly as catalase-negative gram-positive cocci. Growth occurred at 45°C and 60°C for 30 min in broth at pH 9.6 in the presence of 6.5% (w/v) sodium chloride (NaCl) and 40% (w/v) bile, with the reaction on bile esculin agar. The results of physiological and biochemical tests are shown in Table 2. Of the isolated strains, TC4, TC5, and TC6 were similar to *E. faecalis*; TC2 and TC3 were similar to *E. faecium*; and TC1 was an unknown species of *Enterococcus*. The physiological and biochemical characteristics for the unknown *Enterococcus* TC1 were similar to those of the previously described *Enterococcus* species CDC PNS-E2, belong to *E. thailandicus* [2, 11, 13]. Homology analysis by PCR amplification of a 1475-bp fragment of 16S rRNA from the isolated strain TC1 showed consistency with the positive control strip. These results confirmed the assignment of *Enterococcus* species CDC PNS-E2 with a 99% identity (EMBL Nucleotide Sequence Database Accession No: CCUG 47861).

### Virulence factors and antibiotic resistance

The genes *agg*, *cylA*, *efaA*fs, *efaA*fm, *gelE*, *esp*, *ace* and *ptsD* were not detected. However, hemolytic activity was positive (S1 Fig). Analysis of antibiotic susceptibility according to EUCAST (www.eucast.org; version 5.0, January 2015) and results interpreted by EUCAST or by CLSI [16] (Table 3) revealed that that the unknown *Enterococcus* species was susceptible to clinically relevant antibiotics, including ampicillin, chloramphenicol, clarithromycin, gentamicin, norfloxacin, penicillin, tetracycline, and vancomycin. However, it was highly resistant to nitrofurantoin and resistant to moderate levels of ciprofloxacin, and erythromycin.

**Table 2. Physiological characteristics of *Enterococcus* strains isolated from the intestinal microbiota of healthy pigs.**

| Test/characteristic | Strain code | | | | | |
|---|---|---|---|---|---|---|
| | TC1 | TC2 | TC3 | TC4 | TC5 | TC6 |
| Acid production from arabinose | - | + | + | - | - | - |
| Arginine | + | + | + | + | + | + |
| Gas production in MRS broth | - | - | - | - | - | - |
| Glucose | + | + | + | + | + | + |
| Hydrolysis BE | + | + | + | + | + | + |
| Inulin | - | + | + | + | + | + |
| Lactose | + | + | + | + | + | + |
| Mannitol | + | + | + | + | + | + |
| Maltose | + | + | + | + | + | + |
| Melibiose | - | + | + | - | - | - |
| Motility | - | - | - | - | - | - |
| Pyruvate utilization | - | - | - | + | + | + |
| Raffinose | - | - | - | - | - | - |
| Ribose | + | - | - | - | - | - |
| Sucrose | + | + | + | + | + | + |
| Sorbitol | - | - | - | + | + | + |
| Sorbose | - | - | - | - | - | - |
| Trehalose | + | + | + | + | + | + |
| Tolerance to tellurite | - | - | - | + | + | + |
| Xylose | - | - | - | - | - | - |

TC, TongCheng, Hubei; MRS, de Man, Rogosa, and Sharpe; BE, hydrolysis of esculin in the presence of bile; +, positive; -, negative.

## Animal experiments

Observation of general health status revealed no noticeable behavioral or activity changes in the mice, and no treatment-related illness or death occurred. No differences in hair luster and feces were found between the experimental and control groups throughout the experiment. No viable bacteria from the livers, spleens, hearts, or kidneys of any mouse were successfully

**Table 3. Antibiotic resistance of *E. thailandicus* TC1.**

| Antibiotics | Susceptibility tablet dose | Criteria | | | Inhibitory zone diameter (mm) |
|---|---|---|---|---|---|
| | | S | I | R | |
| Ampicillin | 10 µg | ≥17 | - | ≤16 | 23.34 |
| Chloramphenicol | 30 µg | ≥18 | 13–17 | ≤12 | 20.23 |
| Ciprofloxacin | 5 µg | ≥21 | 16–20 | ≤15 | 19.53 |
| Clarithromycin | 15 µg | ≥18 | 14–17 | ≤13 | 18.55 |
| Erythromycin | 15 µg | ≥23 | 14–22 | ≤13 | 15.00 |
| Gentamicin | 120 µg | ≥10 | 7–9 | ≤6 | 19.42 |
| Nitrofurantoin | 300 µg | ≥17 | 15–16 | ≤14 | 13.55 |
| Norfloxacin | 10 µg | ≥17 | 13–16 | ≤12 | 18.20 |
| Penicillin | 10 U | ≥15 | - | ≤14 | 17.39 |
| Tetracycline | 30 µg | ≥19 | 15–18 | ≤14 | 25.83 |
| Vancomycin | 30 µg | ≥17 | 15–16 | ≤14 | 21.25 |

R: resistant; I: intermediary; S: susceptible.

**Table 4. Effects on internal organ indices of mice orally inoculated with *E. thailandicus* TC1 and the control diet.**

| Group | organ indices | | | |
|---|---|---|---|---|
| | Liver | Heart | Kidney | Spleen |
| Treatment | 0.043 ± 0.003 | 0.004 ± 0.000 | 0.012 ± 0.001 | 0.002 ± 0.000 |
| Control | 0.044 ± 0.004 | 0.004 ± 0.001 | 0.013 ± 0.002 | 0.003 ± 0.001 |
| *p*-value | 0.18 | 0.83 | 0.42 | 0.06 |

Values are presented as means ± SD. No significant differences were found between the organ indices from the treated group fed with the isolated strain TC1 and the control group (*P*>0.05).

cultured on MRS agar plates from either the experimental or the control group. In addition, no significant differences were found for the internal organ indices between the experimental group and the control group (Table 4).

From clinical observation, no evident distinctions were found in the shape and size of any of the organs, including the liver, spleen, heart, kidney, ileum, caecum, and colon. Also, no hemorrhage, hyperemia, swelling, or necrosis was evident in any of the organs in the experimental and control groups. However, treatment with the isolated strain TC1 caused damage leading to shortened ileum villi as well as decreased ileum crypt height by various degrees in different animals as viewed under microscopic examination. Histological assays (Table 5) showed that ingestion of the isolated strain TC1 ($1 \times 10^{10}$ cfu/mL) had no effect on villus:crypt ratio, while the villi heights and crypt heights of the ileum were significantly different when compared to those of the control group (*P* = 0.025 and *P* = 0.047).

## Discussion

*Enterococcus* species found as commensals in the gastrointestinal tract have a long history of safety and demonstrable beneficial properties [19]. These species are most commonly used as probiotics in animal feed or added to fermented foods [20, 21]. However, some enterococcal species were a major cause of nosocomial infections related to infections in the urinary tract, blood, intraabdominal cavity, and pelvis [3]. Recent studies showed that some *Enterococcus* species have emerged as important pathogens in animal infections with increased mortality [8–10], which presents a serious detriment to the farming industry. *Enterococcus* CDC PNS-E2 isolation from human clinical specimens was associated with invasive infections in humans, which may pose potential hazards [11]. In 2008, the authors proposed the denomination *E. sanguinicola* to designate the specie CDC PNS-E2. However, another group of investigators named a new species as *E. thailandicus* [14]. *E. sanguinicola* (CDC PNS-E2) and *E. thailandicus* were subsequently recognized as being the same species and the name *E. thailandicus* had priority to be the valid denomination [13]. In the present study, *E. thailandicus* TC1 was obtained from healthy pigs. Our intent was to investigate the security of the isolated strain TC1 to confirm its beneficial properties for development as a new probiotic.

**Table 5. Ileum mucosal architecture measurements of mice fed with *E. thailandicus* TC1 and the control diet.**

| Group | Villi height (μm) | Crypt height (μm) | Villus/crypt ratio |
|---|---|---|---|
| Treatment | 160.42 ± 34.09* | 62.62 ± 11.95* | 2.56 ± 0.29 |
| Control | 232.48 ± 17.24 | 80.68 ± 4.00 | 2.88 ± 0.13 |
| *p*-value | 0.025 | 0.047 | 0.25 |

Data are expressed as mean ± SD.

*$P < 0.05$.

Safety assessment with respect to virulence factors, antibiotic resistance, and animal experimentation was an important phase in the choice of *enterococci* as potential probiotics [19, 22]. Hemolysin was one of the most studied virulence traits in the *Enterococcus* genus [23, 24]. Cytolysin, which carries the bactericidal and hemolytic activity, is the major pathogenic factor of enterococci and is responsible for the main characteristic of enterococcal pathogenicity in clinical tests [24]. In this study, the lack of cytolysin phenotypic and genotypic congruence may be explained by the genome's function as a template: the hemolysin gene was located on the plasmid that was easily lost [25]. Further, the *E. thailandicus* may possess genes orthologous to the cylA gene that share hemolytic function.

Antibiotic resistance genes in *Enterococcus* organisms are often plasmid or transposon related, which presents a risk of horizontal gene transfer [19] between humans and animals. Therefore, antibiotic resistance of enterococci is a major concern in the medical setting and for animal breeding [26]. Our results show that *E. thailandicus* TC1 is susceptible to some clinically relevant antibiotics and, more importantly, is the most sensitive to vancomycin. However, this strain was found to be resistant to nitrofurantoin, cefazolin, and cefalotin. Cefazolin and cefalotin-resistant traits were also found in most of the enterococcal strains isolated from human and pig feces.

*E. thailandicus* TC1 was first obtained in this study from healthy pigs, whose appearance was normal. No differences were observed between the two groups of pigs, and no abnormal reactions in appearance were observed in the mice during the course of the study. Healthy mucosa plays a very important role in intestinal function, to prevent potential pathogens and toxigenic substances from invading systemic tissues or disseminating to extraintestinal organs and tissues [17]. However, effects on the gut mucosa were caused by oral ingestion of the isolated strain *E. thailandicus* TC1: a loss of intestinal tract, shortened length of ileum villi, and decreased height of ileum crypts. From these observations, *E. thailandicus* may present a potential health risk.

In summary, this strain first isolated from the healthy pigs was identified as the *E. thailandicus* TC1. None of the 7 genes of interest was detected, and *E. thailandicus* TC1, except for an inherent resistance to antibiotics, is performance-sensitive to most antibiotics. Moreover, oral ingestion of *E. thailandicus* TC1 had no effect on the general health of mice, bacterial translocation, the internal organ indices, nor histology in animal experiments. However, positive hemolytic activity was observed and the administered strain caused macrophage infiltration in the liver and damage to ileum villi, which led to villi shortening, and decreased ileum crypt height by various degrees in different animals as viewed under microscopic examination. *E. thailandicus* TC1 is likely to harbor potential pathogenicity for animals; however, the specific mechanism is not obvious and must be further investigated.

## Supporting information

**S1 Fig. Haemolytic activity of the isolated strain *E. thailandicus* TC1.** 1: Positive control; 2: The isolated strain; 3: Negative control.
(DOCX)

## Author Contributions

**Conceptualization:** Bei Wu.

**Data curation:** Yue Li.

**Formal analysis:** Xiue Jin.

**Funding acquisition:** Xiliang Wang.

**Investigation:** Xiaoying Wu.

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
