## [Decision Letter · Decision Letter 0]

12 May 2021

PONE-D-21-10442

Identification and safety assessment of Enterococcus species CDC PNS-E2 isolated from a healthy pig

PLOS ONE

Dear Dr. wang,

Thank you for submitting your manuscript to PLOS ONE. After careful consideration, we feel that it has merit but does not fully meet PLOS ONE’s publication criteria as it currently stands. Therefore, we invite you to submit a revised version of the manuscript that addresses the points raised during the review process.

I suggest a thorough revision of the article, especially for the names of bacteria and genes, which must be in italic and perfectly write, and also including the reference section. Figure 1 does not contribute and should be removed in the next version.

We look forward to receiving your revised manuscript.

Kind regards,

Rosa del Campo

Academic Editor

PLOS ONE

Journal Requirements:

2. In your Methods section, please provide additional details regarding the animals used in your study and ensure you have described the source. For more information regarding PLOS' policy on materials sharing and reporting, see https://journals.plos.org/plosone/s/materials-and-software-sharing#loc-sharing-materials.

5. Please include your tables as part of your main manuscript and remove the individual files. Please note that supplementary tables (should remain/ be uploaded) as separate "supporting information" files.

Reviewers' comments:

Reviewer's Responses to Questions

**Comments to the Author**

1. Is the manuscript technically sound, and do the data support the conclusions?

Reviewer #1: Yes

Reviewer #2: Partly

2. Has the statistical analysis been performed appropriately and rigorously? 

Reviewer #1: Yes

Reviewer #2: Yes

3. Have the authors made all data underlying the findings in their manuscript fully available?

Reviewer #1: Yes

Reviewer #2: No

4. Is the manuscript presented in an intelligible fashion and written in standard English?

Reviewer #1: Yes

Reviewer #2: No

5. Review Comments to the Author

Reviewer #1: The article is interesting an well written. The characterization of new species of Enterococcus is important due to their potencial pathogenicity and also their posible importance as probiotics.

The article is sound. I would suggest that the authors look at other virulence factors as those studied are mostly present in E. faecalis. Other facters present in E. faecium could be present. Please take a look at PMID: 29149293.

Reviewer #2: The manuscript describes the characterization of a strain belonging to an uncommon species of Enterococcus isolated from the intestinal microbiota of a healthy pig. Characteristics potentially associated with health risks were investigated. The results suggest that the isolate may be associated with potential health risk and therefore should not be used as a probiotic, at least until further studies are performed.

The work brings a contribution by adding information on the potential health risk represented by the isolate of a rarely found enterococcal species.

However, the manuscript needs an overall revision to adjust some language issues and to clarify some methodological aspects. A few examples are as follows:

1.The title should be changed to accommodate the comments given below.

2. Abstract:

Line 24: It is recommendable to change the statement “Enterococcus has the dual characteristics of…..” for “ Enterococci have the dual characteristics of being …..”

Line 25, 26 and 35, as well as in many other parts of the text. CDC PNS-E2 was described in 2004 (reference 11 of this manuscript), so it is no longer a newly described species. Furthermore, the use of the denomination CDC PNS-E2 is no longer recommended. In 2008, the authors proposed the denomination Enterococcus sanguinicola (reference 2 of this manuscript) to designate the species. However, in another publication released about a few weeks before, another group of investigators named a new species as Enterococcus thailandicus (Reference: Int J Syst Evol Microbiol. 58:1630-4. 2008). E. sanguinicola (CDC PNS-E2) and E. thailandicus were subsequently recognized as being the same species and the name E. thailandicus had priority to be the valid denomination

So the information given in different parts of the present manuscript should be adjusted to this taxonomic issue.

3. Introduction

Line 40: The term “enterococci” should not be in italics and this should be corrected in many other occasions along the text. The same applies to “staphylococci” and “streptococci” (line 65)

Line 43: the statement “Since the enterococci was identified as a separate genus….” Should be changed for “Since the Enterococcus was identified as separate genus”.

Lines 57-66: the contents of this paragraph should also be revised at the light of updated taxonomic information, since CDC PNS-E2 and CDC PNS-E3 have also received valid species denominations.

4.Material and Methods

Lines 81-82. Sentence starting as “ Trace biochemical ……..” is quite confusing. What exactly that means?

Line 88: replaced “Isolation” by “Bacterial isolation”

Line 135: the reference (14) given is related to the evaluation of enterococci with low- and medium-level VanB-type vancomycin resistance. A broader spectrum reference for susceptibility testing, such as EUCAST or CLSI should be provided. In the results section, line 198, the authors mention the CLSI for interpretation, but they do not mention which CLSI document and year they have used. The same document or reference used to interpret is supposed to be used for performing the tests.

Lines 137-139: The list of antibiotics should be in alphabetic order (and the same applies to when they are listed in Table 3). Again, it is important to specify the reference used, because there are different recommendations. For example, 3 cephalosporins were tested in the present study, but the CLSI does not recommend to test chephalosporins against enterococci and therefore does not propose any interpretation. Plase, clarify. Additionaly, CLSI recommends to use gentamicin disks containing 120 ug instead of the 10ug disks used in the present work ….

5. Other sections

Figures 1 and 2 are Ok, but not really necessary

The title of Table 2 should be revised to indicate that it contains the “Physiological characteristics of Enterococcus strains isolated from the intestinal microbiota of healthy pigs”

Additionaly, the identity of TC1-6 should be stated as a footnote for this table

6. PLOS authors have the option to publish the peer review history of their article (what does this mean?). If published, this will include your full peer review and any attached files.

Reviewer #1: **Yes: **Ana P. Tedim

Reviewer #2: No

---

## [Author Response · Author response to Decision Letter 0]

9 Jun 2021

Dear Editor Rosa del Campo,

Thank you very much for your decision letter and advice on our manuscript (Manuscript # PONE-D-21-10442) entitled “Identification and safety assessment of Enterococcus species CDC PNS-E2 isolated from a healthy pig”. We also thank the reviewers for the constructive comments and suggestions. We have revised the manuscript accordingly, and all amendments are indicated by red font in the revised manuscript. In addition, our point-by-point responses to the comments are listed below this letter.

This revised manuscript has been edited and proofread by Medjaden Inc.

We hope that our revised manuscript is now acceptable for publication in your journal and look forward to hearing from you soon. 

With best wishes,

Yours sincerely,

Xiliang Wang

First of all, we would like to express our sincere gratitude to the reviewers for their constructive and positive comments.

Replies to Editor

1. I suggest a thorough revision of the article, especially for the names of bacteria and genes, which must be in italic and perfectly write, and also including the reference section.

Response: Thank you for your suggestion. The article was carefully revised. The names of bacteria and genes were modified in revised manuscript as marked in red color. The reference section was revised as required by PloS One journal’s guideline. 

2. Figure 1 does not contribute and should be removed in the next version.

Response: Figure 1 was removed as you suggested.

3. In your Methods section, please provide additional details regarding the animals used in your study and ensure you have described the source.

Response: The animal details were added in the “Bacterial isolation” section of the Materials and Methods part as follows: Sterile cotton swabs were used to acquire rectal content samples from 100-day-old (N = 13) and 21-day-old healthy Tongcheng pigs (N = 6) at the Tongcheng Reservation Farm and from 40-day-old healthy large white pigs (N = 11) at the Animal Husbandry Co., Ltd. Hubei Three Lake (Hubei, China). Bacterial samples were obtained from totally 30 pigs.

4. PLOS requires an ORCID iD for the corresponding author in Editorial Manager on papers submitted after December 6th, 2016. Please ensure that you have an ORCID iD and that it is validated in Editorial Manager. To do this, go to ‘Update my Information

Response: The corresponding author’s ORCID iD information was updated as required.

5. In your cover letter, please note whether your blot/gel image data are in Supporting Information.

Response: The figures were deleted from the manuscript and was submitted as Supporting data.

Replies to Reviewer 1

Specific Comments

1. The article is sound. I would suggest that the authors look at other virulence factors as those studied are mostly present in E. faecalis. Other facters present in E. faecium could be present. Please take a look at PMID: 29149293.

Response: Thank you for your insightful suggestion. The virulence gene esp was examined in this research as well as in the reference as you mentioned. Furthermore, Enterococci virulence genes for gelatinase (gelE), cytolysin (cylA), adhesion to collagen (ace), cell-wall adhesion (efaA), and cell adhesion (agg) were also investigated in this study. The examination of virulence gene ptsD was added in the revised manuscript.

Replies to Reviewer 2

Specific Comments

1. The title should be changed to accommodate the comments given below.

Response: The title was changed to ‘Identification and safety assessment of Enterococcus thailandicus TC1 isolated from healthy pigs’ in revised manuscript. 

2. Line 24: It is recommendable to change the statement “Enterococcus has the dual characteristics of…..” for “ Enterococci have the dual characteristics of being …..”

Response: We have changed this sentence in Abstract to ‘Enterococci have the dual characteristics of being opportunistic pathogens and promising probiotics.’

3. Line 25, 26 and 35, as well as in many other parts of the text. CDC PNS-E2 was described in 2004 (reference 11 of this manuscript), so it is no longer a newly described species. Furthermore, the use of the denomination CDC PNS-E2 is no longer recommended. In 2008, the authors proposed the denomination Enterococcus sanguinicola (reference 2 of this manuscript) to designate the species. However, in another publication released about a few weeks before, another group of investigators named a new species as Enterococcus thailandicus (Reference: Int J Syst Evol Microbiol. 58:1630-4. 2008). E. sanguinicola (CDC PNS-E2) and E. thailandicus were subsequently recognized as being the same species and the name E. thailandicus had priority to be the valid denomination. So the information given in different parts of the present manuscript should be adjusted to this taxonomic issue.

Response: Thank you for your kind advice. We used E. thailandicus instead of CDC PNS-E2 in this manuscript. 

4. Line 40: The term “enterococci” should not be in italics and this should be corrected in many other occasions along the text. The same applies to “staphylococci” and “streptococci” (line 65)

Response: We have corrected the typefaces of enterococci, staphylococci and streptococci in revised manuscript.

5. Line 43: the statement “Since the enterococci was identified as a separate genus….” Should be changed for “Since the Enterococcus was identified as separate genus”.

Response: The sentence was modified in revised manuscript as mentioned.

6. Lines 57-66: the contents of this paragraph should also be revised at the light of updated taxonomic information, since CDC PNS-E2 and CDC PNS-E3 have also received valid species denominations.

Response: We used E. thailandicus instead of CDC PNS-E2 or CDC PNS-E3 in this manuscript. 

7. Lines 81-82. Sentence starting as “Trace biochemical …….” is quite confusing. What exactly that means?

Line 88: replaced “Isolation” by “Bacterial isolation”

Response: The description of biochemical reagents and antimicrobial agents were updated in revised manuscript as follows:

Biochemical reagents (i.e. arabinose, pyruvate, MRS broth, tellurite, arginine, glucose, inulin, lactose, mannitol, maltose, melibiose, raffinose, ribose, sucrose, sorbitol, sorbose, trehalose, and xylose) and antimicrobial agents Trace biochemical reaction tubes(i.e. vancomycin, tetracycline, ampicillin, chloramphenicol, gentamicin, penicillin, erythromycin, nitrofurantoin, cefazolin, cefoperazone, cefalotin, clarithromycin, norfloxacin, and ciprofloxacin) and susceptibility pieces were obtained from Hangzhou Microbe Reagent Co., Ltd. (Hangzhou, China). 

In addition, we used “Bacterial isolation” instead of “Isolation” in this manuscript.

8. Line 135: the reference (14) given is related to the evaluation of enterococci with low- and medium-level VanB-type vancomycin resistance. A broader spectrum reference for susceptibility testing, such as EUCAST or CLSI should be provided. In the results section, line 198, the authors mention the CLSI for interpretation, but they do not mention which CLSI document and year they have used. The same document or reference used to interpret is supposed to be used for performing the tests.

Response: Thank you for your insightful suggestion. The detailed information for detection of antibiotic resistance was added in revised manuscript as follows:

Antimicrobial susceptibility was tested by the Kirby-Bauer disk diffusion method in M-H agar, as described by the manufacturer, and by broth microdilution according to EUCAST (www.eucast.org; version 5.0, January 2015) and results were interpreted by EUCAST or by CLSI [16]. Staphylococcus ATCC 25923 was used as a reference strain. The following antibiotics (concentrations given in Table 3) were applied in the present study: ampicillin, chloramphenicol, ciprofloxacin, clarithromycin, erythromycin, gentamicin, nitrofurantoin, norfloxacin, penicillin, tetracycline and vancomycin.

9. Lines 137-139: The list of antibiotics should be in alphabetic order (and the same applies to when they are listed in Table 3). Again, it is important to specify the reference used, because there are different recommendations. For example, 3 cephalosporins were tested in the present study, but the CLSI does not recommend to test chephalosporins against enterococci and therefore does not propose any interpretation. Plase, clarify. Additionaly, CLSI recommends to use instead of the 10ug disks used in the present work ….

Response: Thank you for your insightful suggestion. According to CLSI and your suggestion, we deleted the data related to cefazolin, cefoperazone and cefalotin, and added the antibiotic resistance results of 120 ug gentamicin disks. And the antibiotics in Table 3 were listed in alphabetic order in revised manuscript.

10. Figures 1 and 2 are Ok, but not really necessary

Response: Figure 1 was removed and figure 2 was submitted as supporting data.

11. The title of Table 2 should be revised to indicate that it contains the “Physiological characteristics of Enterococcus strains isolated from the intestinal microbiota of healthy pigs”. Additionaly, the identity of TC1-6 should be stated as a footnote for this table

Response: The title of Table 2 was revised to “Physiological characteristics of Enterococcus strains isolated from the intestinal microbiota of healthy pigs”. The Enterococcus strains from TongCheng, Hubei were abbreviated as TC, which was marked as a footnote in Table 2.

---

## [Editor Report · Decision Letter 1]

15 Jun 2021

PONE-D-21-10442R1

Identification and safety assessment of Enterococcus thailandicus TC1 isolated from healthy pigs

PLOS ONE

Dear Dr. wang,

Thank you for submitting your manuscript to PLOS ONE. After careful consideration, we feel that it has merit but does not fully meet PLOS ONE’s publication criteria as it currently stands. Therefore, we invite you to submit a revised version of the manuscript that addresses the points raised during the review process.

Writing the names of the bacteria is as important as being rigorous in the experiments. After the reviewers and myself have pointed out the errors, major mistakes still remain. The first time a microorganism is named, the name should be in full: Enterococcus thailandicus, but from this point on it should be contracted to E. thailandicus, of course in italics.

Other errors are in the word "pstD" in line 31 which is in another style of letter, a space before the bracket in line 62, line 79 aureus should be in italics, and the final points of the statements of the tables.

We look forward to receiving your revised manuscript.

Kind regards,

Rosa del Campo

Academic Editor

PLOS ONE
---

## [Author Response · Author response to Decision Letter 1]

15 Jun 2021

Replies to Editor

1. Writing the names of the bacteria is as important as being rigorous in the experiments. After the reviewers and myself have pointed out the errors, major mistakes still remain. The first time a microorganism is named, the name should be in full: Enterococcus thailandicus, but from this point on it should be contracted to E. thailandicus, of course in italics.

Response: Thank you for your suggestion. The mistakes were modified in revised manuscript as marked in red color. 

2. Other errors are in the word "pstD" in line 31 which is in another style of letter, a space before the bracket in line 62, line 79 aureus should be in italics, and the final points of the statements of the tables.

Response: Thank you for your suggestion. The mistakes were modified in revised manuscript as marked in red color.

---

## [Editor Report · Decision Letter 2]

21 Jun 2021

Identification and safety assessment of Enterococcus thailandicus TC1 isolated from healthy pigs

PONE-D-21-10442R2

Dear Dr. wang,

We’re pleased to inform you that your manuscript has been judged scientifically suitable for publication and will be formally accepted for publication once it meets all outstanding technical requirements.

Kind regards,

Rosa del Campo

Academic Editor

PLOS ONE
---

## [Editor Report · Acceptance letter]

24 Jun 2021

PONE-D-21-10442R2 

Identification and safety assessment of *Enterococcus thailandicus* TC1 isolated from healthy pigs 

Dear Dr. Wang:

I'm pleased to inform you that your manuscript has been deemed suitable for publication in PLOS ONE. Congratulations! Your manuscript is now with our production department. 

Kind regards, 

on behalf of

Dr. Rosa del Campo 

Academic Editor

PLOS ONE